# Dupilumab-Associated Blepharoconjunctivitis: Clinical and Morphological Aspects

**DOI:** 10.3390/biomedicines11123104

**Published:** 2023-11-21

**Authors:** Federica Serino, Valeria Dattilo, Michela Cennamo, Anna Maria Roszkowska, Massimo Gola, Manfredi Magliulo, Elisabetta Magnaterra, Rita Mencucci

**Affiliations:** 1Eye Clinic, Neuromuscular and Sense Organs Department, Careggi University Hospital, 50139 Florence, Italy; federica.serino@unifi.it (F.S.); valeria.dattilo@unifi.it (V.D.); michelacennamo@libero.it (M.C.); 2Department of Neurosciences, Psychology, Drug Research and Child Health, University of Florence, 50019 Florence, Italy; 3Department of Biomedical and Dental Sciences and Morphofunctional Imaging, University of Messina, 98166 Messina, Italy; anna.roszkowska@unime.it; 4Allergological and Pediatric Dermatology Unit, Azienda USL Toscana Centro, 500122 Florence, Italy; massimo.gola@uslcentro.toscana.it (M.G.); manfredi.magliulo@unifi.it (M.M.); elisabetta.magnaterra@unifi.it (E.M.)

**Keywords:** Dupilumab, atopic dermatitis, meibomian glands, cornea

## Abstract

Purpose: To describe the clinical and morphologic changes in the ocular surface microstructure of patients affected with moderate-to-severe Atopic Dermatitis (AD) before and during Dupilumab treatment. Methods: This is a monocentric observational study on thirty-three patients affected with AD before and during Dupilumab treatment. All patients underwent a slit-lamp examination: complete clinical assessment, Break Up Time test (BUT), Schirmer test, and corneal staining grading (Oxford scale) were performed. Meibomian Glands Dysfunction (MGD) evaluation (Meibography), Non-invasive Keratograph Break Up Time test (NIKBUT), Tear Meniscus Height (TMH), and ocular Redness Score (RS) have been investigated using an OCULUS Keratograph. In vivo images of the conjunctiva, cornea, and meibomian glands have been acquired by confocal microscopy. Results: Sixty-six eyes were included in our study: twenty-two eyes of 11 naive patients with indication for treatment but not in therapy yet (Group 1) and forty-four eyes of 22 patients treated with Dupilumab for at least 4 months (subcutaneous administration of 300 mg every 2 weeks) (Group 2). Either patients treated with Dupilumab or naive patients with moderate-to-severe forms of AD had a tear film instability (TBUT and NIKBUT reduced), whereas the quantity of the tear film was overall normal (Schirmer test and TMH), without statistically significant differences between the two groups. When Meibography was performed with the Keratograph, the difference between Group 1 and Group 2 was statistically significant in terms of Meiboscore (*p* = 0.0043 and *p* = 0.0242, respectively), as well as the difference in terms of mean RS. These results paired well with the confocal microscopy results in which we found a decrease in the goblet cell population in the conjunctival epithelium in the treated group (5.2 cells/mm), along with inflammatory cells that were more concentrated around the adenoid lumina of the meibomian glands. Conclusions: In recent years, the use of Dupilumab has been increasing, but mild-to-severe conjunctivitis is a common side effect. Our major results demonstrate a loss of meibomian glands at the Keratograph examination: we can assume a reduced meibum secretion and an evaporative dry eye with MGD. We suggest that the inflammation of the ocular surface may involve not only the cornea and the conjunctiva, but also the meibomian glands, and Dupilumab may play a role. However, the frequency of clear conjunctivitis is not as common as reported in the literature.

## 1. Introduction

Dupilumab, trade name Dupixent (Regeneron & Sanofi, Tarrytown, NY, USA), is a monoclonal antibody that acts as an interleukin-4 (IL4) receptor antagonist, blocking the signaling of IL4 and IL13, and consequently the expression of Th2 cytokines which are primarily involved in allergic disorders. It has been approved for the treatment of moderate-to-severe Atopic Dermatitis (AD) [1].

AD, also known as atopic eczema, is a chronic inflammation involving skin. It is a common disorder: prevalence is around 10–30% in children and up to 10% of adults in industrialized countries [2]. AD has a higher incidence during infancy, but it can occur at any age, with a male predominance. Typical findings are dry skin and eczematous lesions with pruritus. Etiology is complex: both genetic and environmental factors lead to dysregulation of the immune system and epidermidis. Consequently, patients have a defective skin barrier which becomes more susceptible to any environmental irritants. Patients with AD more commonly experience ophthalmic complications than those affected with other allergic conditions. Dermatitis can generally affect periocular skin, causing eczema of the eyelids. Other characteristic ophthalmic alterations are blepharitis, vernal keratoconjunctivitis, ophthalmic Herpes Simplex (HSV) infection, filamentary keratitis, corneal ulcers, keratoconus, glaucoma, and cataracts [3]. 

Dupilumab itself seems to be related to many ocular side effects, which range from more common mild ocular surface disorders to rare severe inflammation requiring discontinuation of the therapy. Conjunctivitis is the most common side effect. LIBERTY AD CHRONOS was the first long-term, phase 3, placebo-controlled, double-blinded study with Dupilumab; they reported an incidence of conjunctivitis (generally referred to as bacterial, viral, allergic, and atopic) higher in patients in therapy with Dupilumab plus topical corticosteroids (19%) than in patients taking a placebo plus topical corticosteroids (8%) [4]. The exact pathogenesis is still uncertain, but the role of IL-4 and IL-13 on goblet cell proliferation, differentiation, and expression of mucin and immunomodulatory genes has been previously reported [5]. Goblet cells are specialized epithelial cells of the mucosal surface, which play a crucial role in decreasing ocular surface inflammation and evaporative tear loss, producing mucins such as mucin 5AC. It is a high-molecular weight epithelial glycoprotein with a lubricant function. Loss of mucins causes increasing frictional resistance between the eyelid and the globe. Goblet cells also secrete anti-microbial proteins and cytokines, with an innate immunity function [6]. The mucous lining also acts as a barrier against microbiomes and organic matter. Dupilumab acts by inhibiting IL-4 and IL-13 signaling, and, therefore, it interferes with ocular immune homeostasis. Data from histological analyses of conjunctival biopsies from patients treated with Dupilumab show a marked decrease in goblet cells with a median density of 3.3 cells/mm, whereas there were an average of 32.3 cells/mm in the untreated group; in addition, a multicellular immune cell stromal infiltrate was found, primarily T cells (CD3+/CD4+) [7]. Multiple factors are associated with the incidence of conjunctivitis. Baseline Atopic Dermatitis severity and prior allergic conjunctivitis history are considered risk factors for the development of Dupilumab-associated conjunctivitis [3]. This conjunctivitis is mild and self-limiting in most cases, but ocular surface and eyelid inflammation could also be very serious, rarely resulting in cicatricial ectropion, symblepharon, and madarosis, up to ankyloblepharon and ocular surface keratinization. These alterations improve after discontinuation of Dupilumab [7].

The purpose of our study is to describe the clinical and morphologic changes in the ocular surface microstructure of patients affected with moderate-to-severe AD before and during Dupilumab treatment. We aim to demonstrate that there are subclinical alterations of meibomian glands and bulbar conjunctiva in patients in therapy with Dupilumab, even if asymptomatic. 

## 2. Materials and Methods

This is a monocentric prospective observational prevalence study on thirty-three patients affected by moderate-to-severe AD, with an Eczema Area and Severity Index (EASI) > 24. The study was conducted at the Eye Clinic, Neuromuscular and Sense Organs Department, Careggi University Hospital (Florence, Italy) from January to June 2023, and was performed according to the current version of the Declaration of Helsinki (52nd WMA General Assembly, Edinburgh, Scotland, UK, October 2000). Patients were referred to us by the Clinic of the Allergological and Pediatric Dermatology Unit, Azienda USL Toscana Centro. All the patients included in the study signed a written informed consent, agreeing to participate. The study was approved by the Careggi University Hospital Research Ethics Board.

Exclusion criteria consisted of previous ocular surgery with disruption of the ocular surface (e.g., previous glaucoma surgery) and ocular surface diseases of other known causes (e.g., Sjögren syndrome, Amiodarone therapy, previous chemical burns…).

All patients underwent complete clinical assessment, including Best-Corrected Visual Acuity (BCVA) evaluation. First, before any other manipulation of the eyelids, TBUT was measured by instilling 1 drop of fluorescein without anesthetic, and then the tear film was evaluated using a broad beam and cobalt blue filter. The punctate staining pattern of the ocular surface, determined at the slit-lamp examination, was graded with the Oxford scale (from 0 to 5). Tear production was assessed with the Schirmer I test: the strip of filter paper was placed in the junction of the middle and later thirds of both lower eyelids, without anesthesia: millimeters of paper wetting were recorded after 5 min. 

MGD evaluation, NIKBUT, TMH, and ocular RS have been investigated using an OCULUS Keratograph (Oculus GmbH, Wetzlar, Germany). Tests were performed firstly in the right eye, and then in the fellow one. If more than one measurement was taken, the mean value was considered. Firstly, the TMH and then the NIKBUT measurement were obtained. The TMH was measured perpendicular to the inferior lid margin at the medium point relative to the pupil center; the software has an integrated rule. The NIKBUT is the time between the last blink and the first rupture of the Placido rings that the Keratograph system project onto the corneal surface. The average time of all breakup events was considered for data analysis. The NIKBUT is a reproducible measure: previous studies proved that it is even more sensitive in measuring tear film stability than the TBUT evaluation on a slip lamp [8]. An infrared diode in the hardware of the Keratograph makes it possible to trans-illuminate the upper and the lower eyelid to observe and evaluate the morphological changes in the meibomian glands. These were graduated with the Jenvis classification and are reported in Table 1 [9]. The upper and lower lid can be recorded separately. 

The Keratograph 5M device has a color camera with the function of measurement of ocular redness. The software of the Keratograph (v2.14r3) detects the thin conjunctival vessels and evaluates the sclera-to-blood ratio determining an automatic classification of the bulbar redness, which is very helpful to overcome the subjectivity with subsequent errors. This was conducted with scans of the exposed bulbar and limbar conjunctiva; the images were analyzed, and the software generated an RS as a continuous variable (scale from 0–4), in which a higher score indicates more severe Bulbar Redness (BR). 

Finally, in vivo images of the conjunctiva, cornea, and meibomian glands were acquired using confocal microscopy after topical anesthetic application (Oxybuprocaine 0.4%) (Heidelberg Retina Tomograph II/Rostock Cornea Module—Heidelberg Engineering GmbH, Heidelberg, Germany).

Statistical analysis was performed using SPSS Statistics (SPSS Inc., Chicago, IL, USA) software for macOS (Version 26.0). Demographic and clinical characteristics of the two treatment groups were compared using a two-tailed Student’s t-test and Wilcoxon rank-sum test, as appropriate for continuous or ordinal variables, or a Chi-square test for dichotomous or categorical variables, with 95% confidence intervals. The chosen level of statistical significance was a *p*-value < 0.05.

## 3. Results

Sixty-six eyes were included in our study: twenty-two eyes of 11 naive patients with indication for treatment but not in therapy yet (Group 1) and forty-four eyes of 22 patients treated with Dupilumab for at least 4 months (subcutaneous administration of 300 mg every 2 weeks) (Group 2). The demographic and clinical characteristics of patients included in the study are summarized in Table 2. 

Among the naive group, three patients (27%) were symptomatic for ocular discomfort. In Group 2, sixteen patients (73%) presented with symptoms of ocular surface disease as itching or dry eye (Figure 1), but only seven patients of sixteen with symptoms (32%) reported worsening conditions after beginning Dupilumab injections (Table 3). No patients have developed sight-threatening complications, such as symblepharon and ocular surface keratinization. None discontinued the therapy because of the ocular side effects.

The punctate staining pattern of the ocular surface was graduated with the Oxford scale (0–5). The mean value was 0.5 in Group 1 and 0.62 in Group 2 without a statistically significant difference. It is interesting to note that among Group 2, only 7 eyes of 44 (16%) had an Oxford grade 2 of punctate epithelial erosions, while none had a greater value.

The TBUT, NIKBUT, Schirmer test, and TMH measurements are reported in Table 4. The results show that there are not statistically significant differences between the two groups (*p* > 0.05).

Figure 2 represents the percentage of eyes for each grade of Jenvis Meiboscore, respectively, for the upper (left) and lower (right) eyelids. The difference between the naive group and the treated group was statistically significant (*p* = 0.0043 and *p* = 0.0242, respectively): that means that among Group 2 more patients experience atrophic changes in meibomian glands in both eyelids (Figure 3). 

Difference in terms of BR was statistically significant (*p* = 0.0053): in fact, mean value was 1.08 in Group 1, while it was 1.5 in Group 2 (Figure 4).

In Group 2, we observed that both eyelids showed mild-to-moderate distorted anatomy at Meibography. At the confocal examination, we observed a decrease in the goblet cell population in the conjunctival epithelium in this group (5.2 cells/mm); in some cases, goblet cells totally disappeared (Figure 5). In tarsal conjunctiva, inflammatory cells were present, and more concentrated around the adenoid lumina of the meibomian glands. We observed several cell infiltrates, composed of a mix of round cells and cells with multilobate nuclei compatible with neutrophils, which were probably a sign of an inflammatory process involving the conjunctiva (Figure 6).

Meibomian glands showed up as hyperreflective, edematous, and with a dilated acinar structure with a blurred lumen contour, that can be referred to as a meibomitis sign.

## 4. Discussion

Meibomian glands play a crucial role in tear production by secreting lipid content. They are hosted in the posterior lamella of both eyelids, and they normally are linear, perpendicular to the lid margin and 3–4 mm length. We mostly observed a loss of meibomian glands at the Keratograph examination. The Keratograph makes it possible to trans-illuminate the upper and the lower eyelids to observe and evaluate the morphological changes in the meibomian glands. First introduced by Arita et al. in 2008, non-contact Meibography is a safe and reproducible examination that allows us to objectively study the morphologic changes that occur in meibomian glands [9]. We observed a statistically significant difference (*p* = 0.0043 and *p* = 0.0242, respectively, for the upper and lower eyelids), in terms of Meiboscore grading between the two groups: patients treated with Dupilumab had dropout, increasing tortuosity, and shortening of the meibomian glands both in upper and lower lids. Dropout means a loss of meibomian glands, unlike atrophy, which means that glands exist but are no longer functioning. Consequently, we can assume a reduced meibum secretion and an evaporative dry eye related with MGD. This is consistent with previous studies where a lack of meibomian glands are also accompanied by damaged meibomian gland function [7]. Meibomian gland dropout was present even in asymptomatic patients with a relatively normal lid margin, as already reported in the literature [10].

The meibomian glands are holocrine sebaceous glands which secrete lipids onto the ocular surface; these lipids constitute the outer layer of tear film and act as a barrier to prevent rapid evaporation of the tear film [11]. The tear film has a role in maintaining the homeostasis of normal ocular flora and contributing to anti-microbial defense. Thus, alterations of the tear film determine bacteria flora changes [12] and a vicious cycle of chronic inflammation. Moreover, it was previously found that Dupilumab interferes with tear levels of mucin 5AC (MUC5AC) by inhibiting IL-13 [13]. This cytokine is directly associated with mucin production. We found that either patients treated with Dupilumab or naive patients with moderate-to-severe forms of AD had a tear film instability since the TBUT and NIKBUT were reduced, whereas the quantity of the tear film, evaluated with the Schirmer test and TMH, was overall normal, without statistically significant differences between the two groups; this is consistent with MGD in both groups. Generally, patients with AD even without Dupilumab treatment have a greater risk of ocular disease, in particular allergic, atopic, and vernal conjunctivitis [14]. However, we expected data of a greater tear film instability in the treated group than in the naive, due to the consistent loss of meibomian glands found at the Keratograph examination. We hypothesized that this finding could be impaired by the greater use of tear film substitutes in the treated group, at the time of our visits. Actually, 85% of the patients were already using prophylactic artificial tears before starting treatment with Dupilumab as prescribed by their dermatologist. There is evidence in the literature that prophylactic use of artificial tears reduces the incidence of ocular surface complications among those being treated with Dupilumab [15,16]. Therefore, this aspect should be further investigated.

In vivo confocal microscopy provides non-invasive high-resolution images of ocular surface tissues, bypassing the need for biopsy and/or impression cytology. Our in vivo images of bulbar and tarsal conjunctiva, acquired by confocal microscopy, confirmed the reduction in intraepithelial goblet cells as reported in the literature in patients treated with Dupilumab (Figure 5) [7]. Goblet cells are round cells, that occur mainly individually or in clusters, and normally account for 10% of the conjunctival epithelial cells [17]. We found a mean value of 5.2 cells/mm. In addition, the in vivo images show inflammatory cells in the tarsal conjunctiva and around the adenoid lumina. We assumed that the inflammation could be the cause behind the dropout of glands. 

We found that patients treated with Dupilumab had red eye more frequently compared to patients not in therapy, yet with a statistically significant difference (*p* = 0.0053). The Keratograph 5M allowed us to objectively quantify BR. Ocular redness is a consistent sign of the ocular response to a pathological stimulus [14]. Dry Eye Disease (DED) is associated with chronic conjunctival hyperemia: the proinflammatory response to the altered ocular environment causes inflammation and consequently conjunctival hyperemia [15]. 

However, atopic blepharitis is one of the major ocular complications of Atopic Dermatitis [16]. We hypothesized that the drug may play a crucial role: a previous study found the presence of Dupilumab in conjunctival cell suspensions, and that drug level was higher in tears of patients with a more severe form of ocular surface disease [18]. Patients affected with AD have dry skin with barrier disruption [16]. We cannot exclude the direct role of the medication on the worsening of the pre-existing blepharitis.

## 5. Conclusions 

In conclusion, our results showed that patients with AD, either naive or treated with Dupilumab, have a tear film instability (TBUT and NIKBUT reduced), whereas the quantity of the tear film is overall normal (Schirmer test and TMH), but without statistically significant differences between the two groups (*p* > 0.05). The major limit of our study is the different sample size between the two groups; however, the statistical analysis showed that the results were significant. 

Our findings also demonstrated a loss of meibomian glands at the Keratograph examination during Dupilumab treatment: we can assume this could be the results of the alteration of the ocular immune homeostasis caused by the inhibition of IL-4 and IL-13. In addition, the in vivo images of the meibomian glands, acquired by confocal microscopy, show inflammatory cells in the tarsal conjunctiva and around the adenoid lumina, and a loss of goblet cells. Our study suggests that the inflammation of the ocular surface may involve not only the cornea and the conjunctiva, but also the meibomian glands, and Dupilumab may play a role. Moreover, conjunctivitis is an unspecific term that simply means inflammation of the conjunctiva. In this sense, we can agree with the fact that Dupilumab, probably through an action on goblet cells in one part and in meibomian glands on the other, can cause chronic inflammation of conjunctiva together with a partial loss of meibomian glands, and both these aspects can contribute to a worsening of dry eye status, already present in AD.

## Figures and Tables

**Figure 1 biomedicines-11-03104-f001:**
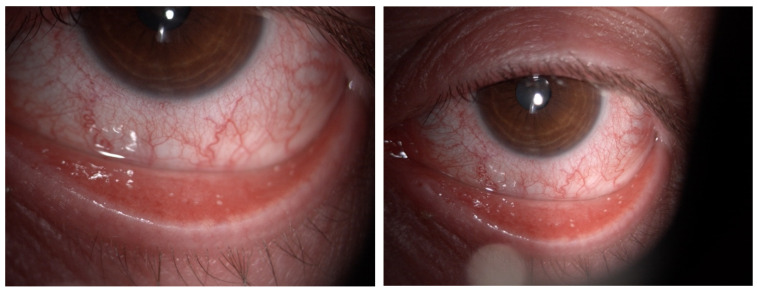
Color images of the eye in a case of Dupilumab-related conjunctivitis in Group 2: four quadrants of bulbar vessels are strongly dilated.

**Figure 2 biomedicines-11-03104-f002:**
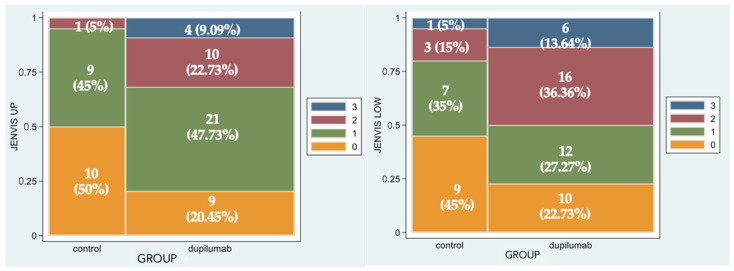
Spine-plot graphical representation of percentage of eyes for each grade of Jenvis Meiboscore of the upper (**left**) and lower (**right**) eyelids. The association between color and meiboscore grading is summarized in the legend.

**Figure 3 biomedicines-11-03104-f003:**
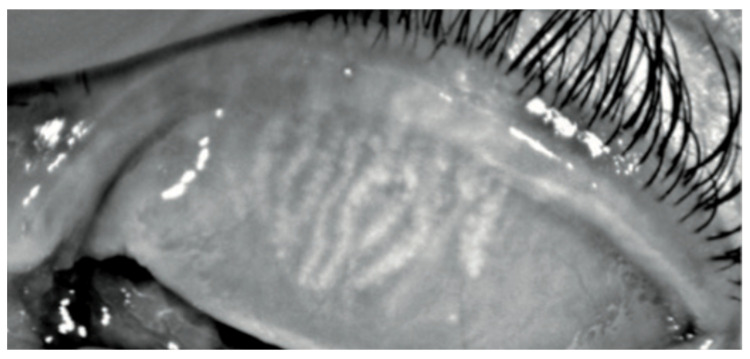
Keratograph 5M Meiboscan acquisition of the upper eyelid with grade 2 (Jenvis): we can observe a dropout of meibomian glands, which appears shorter than normal.

**Figure 4 biomedicines-11-03104-f004:**
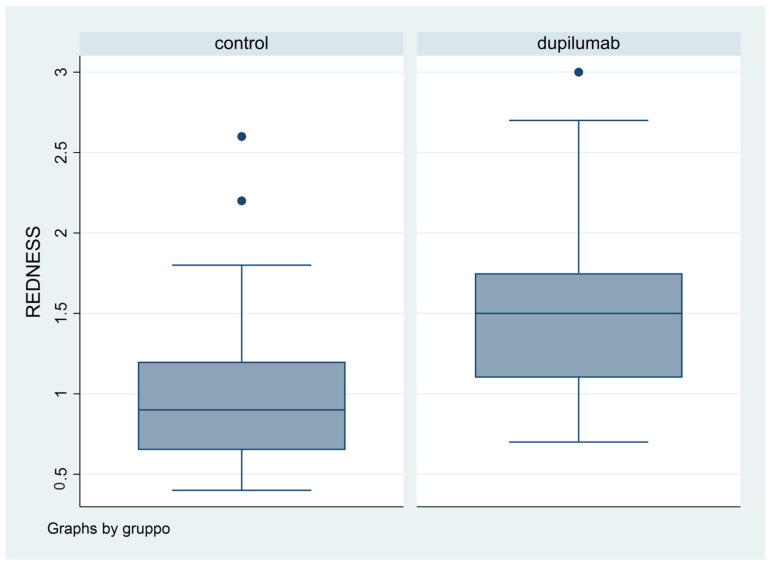
Box plot graphical representation of the Redness Score (RS) for each group.

**Figure 5 biomedicines-11-03104-f005:**
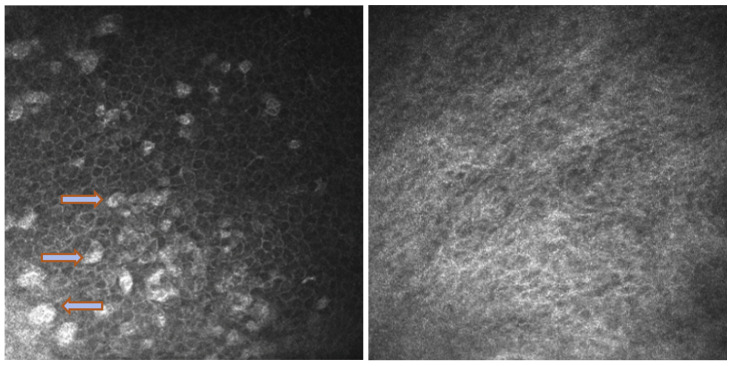
In vivo images of bulbar conjunctiva of a naive patient (**left**) and treated patient (**right**). Note the presence of intraepithelial goblet cells (arrow) on the left, while they are absent in the image on the right.

**Figure 6 biomedicines-11-03104-f006:**
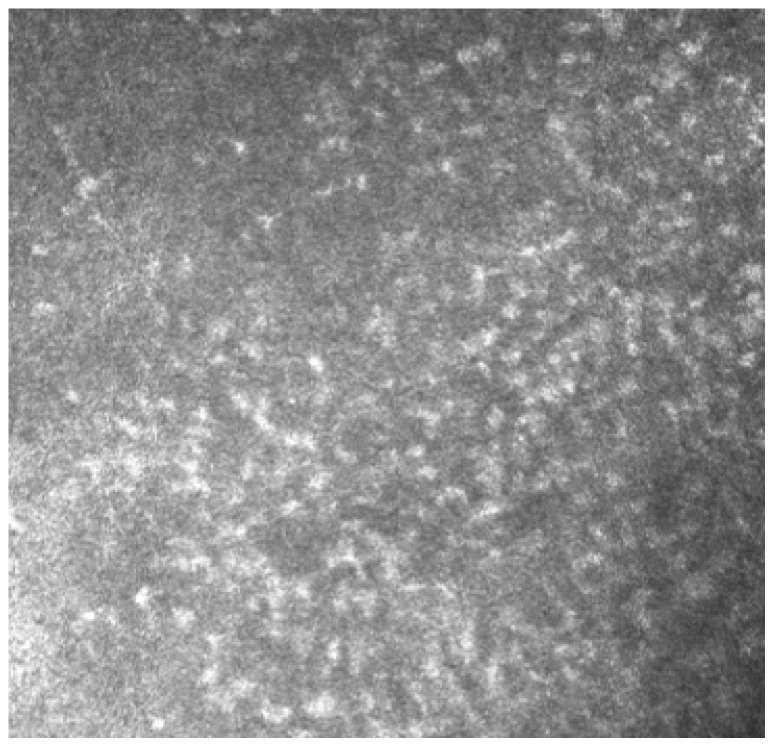
In vivo images of bulbar conjunctiva of a treated patient: several cell infiltrates are present, probably a sign of an inflammatory process involving the conjunctiva.

**Table 1 biomedicines-11-03104-t001:** Meiboscore grading (Jenvis classification) at Keratograph examination: the loss of meibomian glands is graded from 0 to 3.

**Grade 0**	No loss of meibomian glands
**Grande 1**	Loss of less than 1/3 of the total meibomian gland area
**Grade 2**	Loss of 1/3 to 2/3 of the total area
**Grade 3**	Loss of more than 2/3 of the area

**Table 2 biomedicines-11-03104-t002:** Demographic and clinical characteristics of patients included in the study.

	Naive (Group 1)	Treated (Group 2)
**Number of patients (eyes)**	11 (22)	22 (44)
**Age (years), mean value ± SD**	41.0 ± 16.5	45.5 ± 14.3
**Sex**		
Male, n (%)	4 (36%)	12 (54.5%)
Female, n (%)	7 (64%)	10 (45.5%)

Abbreviations: SD: standard deviation; n: number.

**Table 3 biomedicines-11-03104-t003:** Prevalence of referred symptoms at the point of the observation.

	Naive (Group 1)	Treated (Group 2)
Itchy eye	1/11 (9.09%)	6/22 (27%)
Dry eye	2/11 (18.18%)	10/22 (45%)

**Table 4 biomedicines-11-03104-t004:** Results at the point of the observation.

	Naive (Group 1)	Treated (Group 2)	*p*-Value
TBUT (mean value ± SD—s)	8.25	7.18	0.31
NIKBUT (mean value ± SD—s)	13.28	13.85	0.74
Schirmer test (mean value ± SD—mm)	11.6	12.27	0.71
Tear Meniscus Height Normal TMH > 0.20 mm (mean value ± SD—mm)	0.36	0.38	0.74

Abbreviations: SD: standard deviation.

## Data Availability

The data presented in this study are available on request from the corresponding author. The data (original imaging) are not publicly available due to privacy issues.

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
