# Peer review of "Dupilumab-Associated Blepharoconjunctivitis: Clinical and Morphological Aspects"

_biomedicines, 2023, doi:10.3390/biomedicines11123104_

Round 1
Reviewer 1 Report
Comments and Suggestions for Authors
This is a clinical study to compare morphological changes of ocular surface microstructure between the patients treated with dupilumab and those without dupilumab in the patients with moderate to severe atopic dermatitis. The authors found loss of meibomian glands in the some patients with atopic dermatitis. This finding is important to manage the ocular change during the dupilumab treatment for topic dermatitis. There are several points to be clarified.
Major points
Point 1 In the Introduction section: The authors should clearly state what hypothesis you want to prove in this clinical study.
Point 2 In Table 2: Was the result obtained at the starting point of the observation ? If it is so, this should be described as “Table 2 Prevalence of referred symptoms at the starting point of the observation.”
Point 3 In Table 3: What timing were these results obtained, at the starting point or observation periods ? This information should be presented in the legend of Table 3.
Point 4 In Table 4: Two tables are presented, however no clear explanation was presented, either at the starting points or at the observation period.
Point 5 In Table 5: The information for what timing was the result obtained either at the starting points or at the observation period.
Point 6 In Discussion (Page 6 line 161-175): Answer of the purpose of the study should be presented at the beginning of the Discussion section. Then, the basis for the answer should be explained precisely. Thereby the description “The employment of Dupilumab is …… primarily comprised of T cells (CD3+/CD4+) [7]” should be moved to Introduction section.
Comments on the Quality of English LanguageMinor points
Point 1 Page 1 line 17 and line 25: BUT and TBUT should be clearly explained.
Point 2 Page 1 line 29: “as well as well as” should be “as well as”.
Points3 Page 2 line 47-48: “Patients with AD more commonly … other allergic conditions.” This sentence needs reference.
Point 4 Page 2 line 95: TREATD may be treated.
Point 5 Table 4: Gruppo may be groups.
Point 6 Page 5 line 146: “globet cells” may be “goblet cells”. The description of globet cells appears many times the after.
Author Response
Thank you very much for the review of our manuscript. We sincerely appreciate all valuable comments and suggestions, which helped us to improve the quality of the article. Our responses are described below in a point-to-point manner.
MAJOR POINTS
- “We aim to demonstrate that there are subclinical alterations of meibomian glands and bulbar conjunctiva in patients in therapy with Dupilumab, even if asymptomatic”. This specification has been added in the text.
- We reworded “Prevalence of referred symptoms at the point of the observation”. Since this is a prevalence study, there is just one point of observation.
- We reworded “Results at the point of the observation”. Since this is a prevalence study, there is just one point of observation.
- We added an explanation of Table 4. We reworded “ Meiboscore grading (Jenvis classification) at Keratograph examination: the loss of meibomian glands is graded from 0 to 3”.
- As regards Figure 2, the two plot are referred respectively to the UPPER AND LOWER EYELID. We improved the legend.
- There is not a Table 5 in our manuscript. Do you mean Figure 5?
- We moved the section to the Introduction, and we answered the question at the beginning as you suggested.
MINOR POINTS
- Abbreviations have been clarified.
- Done
- Done
- Done
- Done
- Done

Reviewer 2 Report
Comments and Suggestions for Authors
This is a good manuscript that addresses an ocular manifestation of Dupilumab, however, the style of presentation requires major modification. A significant number of sentences need to be reworded to render them more comprehensible. The authors clearly stated the purpose of the work, and the introduction provided relevant background information about the topic being discussed. The description of the methodology is satisfactory. The authors should ensure that all abbreviations used in the manuscript for the first time are written in full. I would like the authors to address these suggestions/comments raised by the reviewer.
Specific suggestions/comments raised by the reviewer:
Lines 16 – 17: The sentence should be reworded for better clarity.
Line 28: Render “significant different” as “significant”
Lin e 44: Render “IL-23” as “IL-13”
Lines 55 – 57: The sentence should be reworded for better clarity.
Line 88: “… using a two-tailed Student’s t-test or Wilcoxon rank-sum test…” These two statistical tests are usually complementary and not alternatives. Is it “or” or “and”?
Line 111: “… among group 2 only 7 eyes (16%) had an OXFORD grade 2 of ... greater value”: The authors need to be more explicit with numbers here; e.g. “… only 7 eyes (16%) out of ??? eyes …” (??? needs to be specified as the case may be).
Table 2: Verify the accuracy of the contents of the table. There is no reported row for “punctate epithelial erosions”. See line 111.
Table 3: Is “DS” an abbreviation for standard deviation?
Line 122: Render “significant different” as “significant”
Line 165: Render “ocular inflammation” as “ocular surface inflammation”
Line 165: Render “acts” as “acts by”
Line 169: Render “addiction” as “addition”
Lines 194 – 196: The sentence should be reworded for better clarity.
Line 215: Render “limit” as “limitation” I doubt if the major limitation stated by the authors is a concern. If the difference in sample size between the two groups is a concern, please provide a power calculation
Comments on the Quality of English LanguageA significant number of sentences need to be reworded to render them more comprehensible.
Author Response
Thank you for your helpful suggestions and positive comments. Our responses are described below in a point-to-point manner.
- We reworded: All patients underwent a slit lamp examination: complete clinical assessment, Break Up Time test (BUT), Schirmer test, corneal staining grading (Oxford scale) were performed.
- Done
- Done
- We reworded: they reported an incidence of conjunctivitis (generally referred as bacterial, viral, allergic, and atopic) higher in patients in therapy with Dupilumab plus topical corticosteroids (19%) than in patients taking placebo plus topical corticosteroids (8%) [3]
- “And”. We changed.
- We reworded: . It is interesting to note that among group 2 only 7 eyes of 44 (16%) had an OXFORD grade 2 of punctate epithelial erosions, while none had greater value.
- There isn’t a row because punctate erosion is a sign, while in the table we reported the referred symptoms.
- Done
- Done
- Done
- Done
- Done
- We reworded “Actually, 85% of the patients were already using prophylactic artificial tears before starting treatment with Dupilumab as prescribed by their dermatologist”
- The therapeutic indication of Dupilumab is specific. We decided to enroll all patients with moderate-to severe AD which have been referred to us for 6 months (from March to August 2023).

Round 2
Reviewer 1 Report
Comments and Suggestions for Authors
The revised manuscript was adequately corrected according the comments.
Reviewer 2 Report
Comments and Suggestions for Authors
The authors have provided an improved version of the original manuscript that highlighted one of the ocular side effects associated with the use of Dupilumab, a monoclonal antibody that inhibits signalling of Th2-derived cytokines (IL-4 and IL-13) through IL-4Ralpha. The authors have provided a good insight on how this biologic agent can hinder the ability of the meibomian gland to produce the lipid layer of the tear film. An absent/low lipid layer can result in tear film hyperosmolarity due to excessive evaporation of the tear film. Evaporative dry eye is a global ocular problem of great concern for eye care professionals and the authors have provided a good insight on another therapeutic agent that can cause evaporative dry eye.